# A metabolic atlas of the *Klebsiella pneumoniae* species complex reveals lineage-specific metabolism and capacity for intra-species co-operation

Ben Vezina[1,2*], Helena B. Cooper[1,2], Christopher K. Barlow[3], Martin Rethoret-Pasty[4], Sylvain Brisse[4], Jonathan M. Monk[5], Kathryn E. Holt[1,6], Kelly L. Wyres[1,2,6*]

**1** Department of Infectious Diseases, School of Translational Medicine, Monash University, Melbourne, Victoria, Australia, **2** Centre to Impact AMR, Monash University, Clayton, Victoria, Australia, **3** Monash Proteomics and Metabolomics Platform, Department of Biochemistry and Molecular Biology, Biomedicine Discovery Institute, Monash University, Melbourne, Victoria, Australia, **4** Institut Pasteur, Université Paris Cité, Biodiversity and Epidemiology of Bacterial Pathogens, Paris, France, **5** Department of Medicine, University of California, San Diego School of Medicine, San Diego, California, United States of America, **6** Department of Infection Biology, London School of Hygiene and Tropical Medicine, London, United Kingdom

* benjamin.vezina@monash.edu (BV); Kelly.wyres@monash.edu (KLW)

## Abstract

The *Klebsiella pneumoniae* species complex inhabits a wide variety of hosts and environments, and is a major cause of antimicrobial resistant infections. Genomics has revealed the population comprises multiple species/sub-species and hundreds of distinct co-circulating sub-lineage (SLs) that are associated with distinct gene complements. A substantial fraction of the pan-genome is predicted to be involved in metabolic functions and hence these data are consistent with metabolic differentiation at the SL level. However, this has so far remained unsubstantiated because in the past it was not possible to explore metabolic variation at scale. Here, we used a combination of comparative genomics and high-throughput genome-scale metabolic modeling to systematically explore metabolic diversity across the *K. pneumoniae* species complex ($n = 7,835$ genomes). We simulated growth outcomes for each isolate using carbon, nitrogen, phosphorus, and sulfur sources under aerobic and anaerobic conditions ($n = 1,278$ conditions per isolate). We showed that the distributions of metabolic genes and growth capabilities are structured in the population, and confirmed that SLs exhibit unique metabolic profiles. In vitro co-culture experiments demonstrated reciprocal commensalistic cross-feeding between SLs, effectively extending the range of conditions supporting individual growth. We propose that these substrate specializations may promote the existence and persistence of co-circulating SLs by reducing nutrient competition and facilitating commensal interactions. Our findings have implications for understanding the eco-evolutionary dynamics of *K. pneumoniae*

**Data availability statement:** All genomic data used is publicly available—accessions and genomes used can be found in S1 Data. Raw metabolomics files are found at https://doi.org/10.5281/zenodo.17705463. All other raw data can be found in S2 Data–S6 Data. Phylogenetic trees, R, and MICOM code can be found at Figshare (https://doi.org/10.6084/m9.figshare.24503737). Interactive phylogeny can be found at https://microreact.org/project/pcTMwQZAGCsqqhbEZzaj11-a-metabolic-at-las-of-the-klebsiella-pneumoniae-species-complex.

**Funding:** This work was funded by the Australian Research Council Discovery Project (DP200103364, awarded to K.L.W., K.E.H., S.B., and J.M.M.). Additionally, K.L.W. was supported by a National Health and Medical Research Council of Australia Investigator Grant (APP1176192). The funders had no role in study design, data collection and analysis, decision to publish, or preparation of the manuscript.

**Competing interests:** I have read the journal's policy and the authors of this manuscript have the following competing interests: K.E.H. is a member of PLOS Biology's Editorial Board. The other authors declare that no competing interests exist.

**Abbreviations:** ANI, average nucleotide identity; COG, clusters of orthologous genes; GFA, graphical fragment assembly; HILIC, hydrophilic interaction liquid chromatography; HRMS, high-resolution mass spectrometry; *Kp*SC, Klebsiella pneumoniae species complex; LC-MS, Liquid Chromatography-Mass Spectrometry; LIN, life identifier number; SL, sub-lineages; STs, sequence types.

and for the design of novel strategies to prevent opportunistic infections caused by this World Health Organization priority antimicrobial resistant pathogen.

## Introduction

*Klebsiella pneumoniae* is a ubiquitous gram-negative bacterium and major cause of opportunistic healthcare-associated infections with significant global burden [1]. Novel prevention and control strategies are urgently required to target this organism, in particular the growing numbers of multi-drug resistant strains that cause infections that are extremely difficult to treat [2]. However, our capacity to design effective control measures is complicated by the extraordinary diversity among *K. pneumoniae* and gaps in our knowledge about how key types of variation, such as metabolic variation, intersect with ecology, and population structure.

Isolates identified as *K. pneumoniae* in clinical laboratories include *K. pneumoniae* sensu *stricto* in addition to six other taxa (species and sub-species) collectively known as the *K. pneumoniae* species complex (*Kp*SC) [3]. These species are separated by 3%–4% nucleotide divergence in their core genes, and each is further subdivided into hundreds or thousands of phylogenetically distinct sub-lineage (SLs) separated by ~0.5% nucleotide divergence [3,4], but with access to a highly diverse shared gene pool. It is well known that gene content differences play a role in driving differential SL epidemiologies, i.e., where a subset is recognized as globally-distributed multi-drug resistant agents of healthcare-associated infections or as 'hypervirulent' community-acquired pathogens (usually drug-susceptible) [3]. It has also been shown that total gene content is nonrandomly distributed between SLs and that up to 37% of the total *Kp*SC gene pool encodes proteins contributing to cellular metabolism [4].

We therefore hypothesized that SLs differ in terms of metabolic capabilities, which likely contributes to their epidemiological behaviors and may facilitate the maintenance of diverse SLs within the population. A similar hypothesis was proposed for the gram-positive bacterial pathogen, *Streptococcus pneumoniae* [5], and is consistent with genomic studies of other organisms that have implicated a nonrandom distribution of metabolic traits within the population [6–10]. However, the complexities of cellular metabolism make systematic prediction of phenotypes from genome data difficult, limiting capacity for large-scale analyses of population metabolism.

Genome-scale metabolic models attempt to encapsulate the total metabolic potential of an organism and can be used to predict metabolic phenotypes [11]. Comparative metabolic modeling analyses can be utilized to probe diversity within bacterial species populations, and recent larger-scale studies ($n = 50$–3,083 genomes) have highlighted the power of these approaches to identify industrially relevant, ecology-, drug-resistance- and pathogenicity- associated metabolic traits [12–18]. While a recent study of 2,773 metabolic models confirmed that *Escherichia coli* phylogroups (which are approximately as diverse as *Kp*SC species) are associated with unique metabolic reactions [19]. However, none of these works have attempted to investigate the interplay between phenotypic diversity and fine-scale population structure, i.e., to understand how diversity is distributed within and between phylogenetic SLs.

Such information is crucial to understanding the interaction between metabolism, ecology and evolution within species, and in turn for design of broad-acting control strategies that seek to exploit metabolic vulnerabilities.

Here, we describe a population-level analysis of 7,835 *Kp*SC genomes to evaluate the distribution and diversity of metabolism in the context of the population structure, first at the level of species, then SLs within species. We performed quantitative analyses of metabolic gene content, and constructed metabolic models from which we predicted 1,278 metabolic phenotypes for each isolate. This approach allowed us to directly infer the phenotypic impact of genetic variation and explore phenotype variation within a single species in the context of the established *Kp*SC population genomics framework [3] and at a scale that was not previously possible. We validated a subset of these phenotypes via experimental assays. Our data reveal considerable variation between and within species that is nonrandomly distributed across SLs. Notably, we show for the first time that *Kp*SC SLs, including multi-drug resistant and hypervirulent clones, are associated with distinct substrate usage profiles that appear to facilitate commensal interactions and which we hypothesize may promote co-existence.

## Materials and methods

### Genome acquisition, quality filtering, and genotyping

Genomes were included from 29 medium-large-scale studies [4,20–46] that comprised diverse *Kp*SC collections (S1 Data). Outbreak investigations were explicitly excluded to reduce clonal oversampling and genomes were de-replicated to minimize sampling biases for overrepresented SLs (see method: Genome dereplication). Reads were acquired from the European Nucleotide Archive using the enaDataGet and enaGroupGet scripts from enaBrowserTools v1.6 [47], trimmed with Trim Galore v0.5.0 using -q 20 option [48], then assembled with Unicycler v0.4.7 with the --keep 0 option [49]. Assemblies failing the established KlebNET-Genomic Surveillance Platform inclusion criteria were excluded (>500 contigs, total length <4,969,898 or >6,132,846 bp). Graphical Fragment Assembly (GFA) files were analyzed to count GFA dead ends using getUnicyclerGraphStats.py from Unicycler [49]

Assemblies were annotated with Bakta v1.1.1 [50] with the '--gram –' option and Bakta database (date accessed: 1/09/2021), while Kleborate v2.0.4 [51] was used to identify species and Sequence Types (STs). Pathogenwatch (https://pathogen.watch/) and BIGSdb (https://bigsdb.pasteur.fr/) [52,53] were used to assign LIN codes [54] and SLs. LIN codes are assigned using core genome MLST profiles that consider variation in 629 core *Kp*SC genes. The third position in the LIN code is used to infer SL membership [54] and results in groupings that correspond to well-recognized deep-branching monophyletic clades within species. Notably, these SL assignments do not rely on de novo cluster analysis, and are stable with the addition of new genomes [55]. Pairwise ANI was calculated using FastANI v1.33 [56] using --fragLen 3000.

### Genome dereplication

We noticed that several SLs were vastly overrepresented in the dataset (e.g., due to dense sampling in specific geographies or inclusion of transmission clusters within larger collections). In order to minimize the impact of these sampling biases we de-replicated the dataset using a similar approach to that described in [51]. Initially, popPUNK v2.4.0 [57] was used to finely cluster isolates. The 'create-db' function was used with the following options: '--sketch-size 1000000 --min-k 15 --max-k 29 --qc-filter prune'. Then the 'fit-model' function was used with the following options: 'dbscan --ranks 1,2,3,5 --graph-weights'. Finally, the 'fit-model' function was used again, with the options: 'refine --graph-weights --unconstrained'. Additionally, the 'poppunk_visualise' function was used, with the '--distances' and '--previous-clustering' utilizing the refined model fit, to output a neighbor-joining core tree.

Isolates which shared the same popPUNK cluster, isolation source and country were sorted using newfangled_untangler.py v1.0.4 (https://github.com/bananabenana/newfangled_untangler/) then de-replicated via Assembly-Dereplicator v0.1.0 [58] with the following options: '--threshold 0.0003'. Code also available at Figshare (https://doi.org/10.6084/m9.figshare.24503737).

## Pan-genome analysis

The 7,835 filtered and dereplicated KpSC genomes were split into species, then Panaroo v1.2.8 [59] was run with the following options: '--clean-mode sensitive -a core --aligner mafft --no_clean_edges --core_threshold 0.98 --merge_paralogs --remove-invalid-genes'. Due to computational limits, the 6,676 *K. pneumoniae* genomes were split into two groups of 3,338 (based on closest genetic distance) and run separately on Panaroo. The output graphs for all species were then merged using the panaroo-merge command with the following options: '--merge_paralogs' to obtain the final pan-genome.

## Metabolic gene identification and analysis

To analyze metabolic pathways, the pan_genome_reference.fa file from Panaroo was translated using the transeq tool from EMBOSS v6.6.0 [60] with the following options: "-table 11 -frame 1." Translated sequences were analyzed using kofam_scan v1.3.0 (https://github.com/takaram/kofam_scan), the command line version of KEGG's [61] kofamKOALA [62]. KEGG release 94.0 ver. 2020-04-02.

Genes with an e-value of ≤0.001 were retained and analyzed using KEGG Mapper (https://www.genome.jp/kegg/mapper/reconstruct.html) to identify pathway information. Using the BRITE tab, genes which fell under the 'metabolism' subheading were used to classify genes as metabolic or nonmetabolic. KEGG ortholog assignments were used to convert the Panaroo gene presence/absence matrix into a metabolic ortholog matrix. This allowed grouping of functional isozymes which would otherwise have an identical KEGG function. Pairwise metabolic ortholog Jaccard similarities were calculated using the vegdist function from R package vegan v2.5-7 [63].

Twilight [64] was used to classify distribution of metabolic orthologs among KpSC taxa and 48 common *K. pneumoniae* SLs ($n \geq 15$ genomes each, defined by LIN codes) using the -s 5 and -s 20 commands, respectively. 'Core': ≥95% prevalence. 'Intermediate': <95% and >15% prevalence. 'Rare': ≤15% and >0% prevalence. 'Absent': 0% prevalence.

## Metabolic modeling

Metabolic models were generated via Bactabolize v1.0.1 [65] using the *draft_model* command with the following options: "--media_type m9 --atmosphere_type aerobic --min_coverage 25 --min_pident 80." The KpSC-pan v2.0 model [66] was used as the input reference. One-hundred and sixty of 7,835 models (2.0%) required gap-filling to enable simulation of growth on M9 minimal media with glucose (median: 1 reaction added, range: 1–23), using flux balance analysis to optimize the most recent version of the biomass objective function "BIOMASS_Core_Feb2022." We have previously shown that models gap-filled to simulate growth in minimal media plus glucose can accurately predict a range of other substrate growth phenotypes [65].

Flux balance analysis was used to predict growth phenotypes for all possible carbon, nitrogen, phosphorus, and sulfur sources supported by the reference model as sole sources of carbon, nitrogen, phosphorus, or sulfur in M9 minimal media defined in silico, in both aerobic and anaerobic conditions. These analyses were performed using Bactabolize's *fba* command with the following option: "--fba_spec_name m9." All growth lower bounds and default substrate sources used in this analysis are available at https://github.com/kelwyres/Bactabolize. Models were considered capable of simulating growth when the optimized biomass value was ≥0.0001.

Pairwise Jaccard similarities and population distribution of positive growth phenotypes were calculated as described above for metabolic orthologs.

To evaluate co-occurrence of metabolic traits and control for population structure, genomes were randomly subsampled to a maximum of 10 per SL ($n = 2,427$ genomes), then predicted aerobic growth phenotypes analyzed using Coinfinder v1.2.0 [67] with the --associate option.

## In vitro growth experiments

To validate substrate usage predictions from the metabolic models, 13 isolates (S2 Data) were selected from our in-house collections [4,26,68] and tested in triplicate on each of nine substrates in minimal media, aerobic conditions. Two

experiments were performed: i) endpoint $OD_{600}$ at 24- and 48-hours; ii) 3-day substrate coaxing to induce gene expression [69] for predicted false-negative growth results in the endpoint experiment. Isolates were coaxed onto the testing substrate via sub-culturing into progressively lower amounts of D-glycerol, which all isolates can use as a carbon source (10 mM on subculture 1, 5 mM on subculture 2, 0 mM on subculture 3).

Nine growth substrates were tested: seven as sole sources of carbon (methanol, L-hydroxyproline, xylitol, butyrate, acetoacetate, D-glycerol, and D-glucose), one as a sole source of nitrogen (allantoin) and one as a sole source of sulfur (methanesulfonate).

For endpoint experiments, isolates were grown overnight in 2 mL M9 media + 10 mM D-glycerol. M9 media was prepared by using 1X M9 Minimal Salts (Sigma) and adding 2 mM $MgSO_4$ and 0.1 mM $CaCl_2$ after autoclaving. 1 mL of culture was pelleted at 8,000 x $g$ for 10 min, supernatant discarded, then centrifugally washed using 1 mL 0.9% NaCl to remove excess carbon/nitrogen/sulfur. Isolates were then diluted 1:100 using 0.9% NaCl and 5 μL sub-cultured into 96-well plates (CLS3603, Corning) containing 200 μL M9 + substrate. Substrates were tested at the following concentrations: 400 mM methanol, 20 mM L-hydroxyproline, 20 mM xylitol, 20 mM sodium butyrate, 20 mM acetoacetate, 20 mM D-glucose, 10 mM D-glycerol 20 mM methanesulfonate, 60 mM allantoin, all at pH 7.0 and 0.2 μm filter sterilized. For allantoin as a nitrogen source, custom nitrogen-negative M9 was prepared (34 mM $NaH_2PO_4$, 64 mM $K_2HPO_4$, 1 μM $FeSO_4$, 2 mM $MgSO_4$, and 0.1 mM $CaCl_2$). For methanesulfonate as a sulfur source, sulfur-negative M9 was prepared (2 mM $MgCl_2$ instead of $MgSO_4$). Isolate negative and carbon positive controls were performed for each media. Plates were incubated at 37 °C, shaking at 150 RPM and optical density read every 24 hours at $OD_{600}$ absorbance using an infinite 200Pro and i-control v2.0.10.0 (Tecan). Each experiment was performed in biological triplicate. Each well was blanked with a no-isolate control, then the mean used to determine growth, with a cutoff of 0.05 (limit of detection).

Coaxing experiments were performed as above, over 3 days. Isolates were added to media containing the tested substrate plus D-glycerol 10 mM on subculture day 1, 5 mM on subculture day 2, and 0 mM on subculture day 3. Prior to sub-culturing, cells were centrifugally washed as above and 5 μL transferred to 200 μL total media. Only subcultures on day 3 were used for analysis (0 mM D-glycerol).

### Co-culture experiments

We selected three pairs of SLs and three corresponding pairs of substrates (four total substrates) for which one was predicted as a SL-specific core trait in SL A and the other predicted as a SL-specific core trait in SL B. For each SL, three isolates were selected from our in-house collections [4,26,70] and treated as biological replicates (S3 Data).

Isolate pairs were grown in single and co-culture using 0.4 μm pore Transwells (Sigma) and 24-well plates (Costar, Corning). Co-culture results were compared to single cultures and $log_2$-fold-change $OD_{600}$ values calculated.

To rule out strain-strain antagonism prior to co-culturing, a primary inoculum was streaked onto LB agar plates and grown for 8 hours at 37 °C. Then, primary inoculums of additional isolates were streaked perpendicularly. Plates were incubated for 16 hours at 37 °C and no growth inhibition of the perpendicular streaks was observed.

Co-culture results were compared to single cultures using $OD_{600}$ of pairs of isolates (S3 Data) that were physically separated by 0.4 μm pores using 24-well Transwells (Sigma). All cultures were grown at 37 °C, 100 RPM. Substrates were used in 1X M9 media at the following concentrations: 20 mM for galactitol, L-sorbose, L-tartrate, and 5 mM for L-hydroxyproline.

M9 media plus the tested substrate was added to the Transwell (100 μL) and the main well (800 μL). Isolates were grown overnight in 1X M9 containing 10 mM D-glycerol, then centrifugally washed as above. Cultures were diluted 1:100 in 0.9% NaCl, and 5 μL added to wells. The prototroph was added to the Transwell while auxotroph added to the main well. Cultures were grown and 75 μL was taken from wells at time point for $OD_{600}$ measurements. For isolate-substrate combinations where we observed slow growth rates, 24, 48, and 72-hour time points were used, otherwise time points at 0, 24, and 36 hours were used. $OD_{600}$ values were blanked using no-isolate media controls. For $log_2$ fold-change analysis,

values of 0 were transformed to $1\times10^{-5}$, but kept raw for other statistical analyses. After co-culture, isolates were then plated onto LB agar, confirmed uniform, distinct colony morphology, then re-grown under the same nutrient conditions in single culture.

## Co-culture simulations

In silico co-culture experiments were performed using MICOM v0.32.2 [71] with the relevant *K. pneumoniae* metabolic models. Community models were built using the build function, then communities were grown in growth media matching co-culture experiments (M9 minimal media with various carbon sources) with tradeoff set to 0.5. Metabolite and reaction flux were then analyzed to determine metabolite sharing from prototrophs to auxotrophs.

## Preparation of samples for metabolomics analysis

We used an untargeted Liquid Chromatography-Mass Spectrometry (LC-MS) approach to explore the metabolites present in the culture supernatant of putative cross-feeding prototrophs. We selected one representative prototroph for each SL-growth substrate combination: AJ155, INF359, INF120 (galactitol), INF171 (L-hydroxyproline), INF225 (L-sorbose), and INF354 (L-tartrate) and compared to no-isolate controls for each substrate. Briefly, overnight cultures in M9 minimal media + 20 mM glucose were washed twice in 0.9% NaCl, centrifuged at 8,000 x *g* for 10 min. Cultures were then standardized to McFarland Standard 0.45–0.55 in 0.9% NaCl, then inoculated into 5 mL M9 minimal media + substrates (as previously described in co-culture experiments). Performed with four biological replicates except negative controls which were performed in triplicate. Independent cultures were grown aerobically at 37 °C, shaking at 200 RPM for 18 hours. Cells were pelleted at 8,000 x *g* for 10 min, then supernatant filtered through a Acrodisc Supor PES Membrane 0.2 μm syringe filter (Pall). Supernatant was transferred to a new vial and frozen at −80 °C prior to transfer to the Monash Proteomics and Metabolomics Platform for LC-MS analysis, where 20 μL of cell culture supernatant was treated with 180 μL of ice-cold methanol containing a mixture standard isotope labeled amino acids acting as internal standards. The sample was then mixed at 4 °C for 10 min on a thermomixer before being subjected to centrifugation at 21,000 x *g* for 10 min at 4 °C. 900 μL of the supernatant was then transferred to an LC-MS vial for analysis.

## LC-MS analysis of metabolites

We used both hydrophilic interaction liquid chromatography (HILIC) coupled to high-resolution mass spectrometry (HRMS). Samples were analyzed on a Vanquish Horizon coupled to Exploris 480 orbitrap mass spectrometer (Thermo Scientific). Chromatography was performed using an iHILIC-(P) Classic, (PEEK, 150 x 4.6 mm, 5μm, 200Å; HILICON, Sweden) with a gradient elution of 20 mM ammonium carbonate (A) and acetonitrile (B) (linear gradient time-%B as follows: 0 min-80%, 15 min-50%, 18 min-5%, 21 min-5%, 24 min-80%, 32 min-80%) at a flow rate of 500 μL/min. The mass spectrometer was operated in polarity switching modes with a resolution at 120,000. The electrospray conditions were: ionization voltage was 3.5 kV in positive and −2.5 kV in negative mode, ion transfer tube temperature = 325 °C; sheath gas = 60; Aux gas = 15; sweep gas = 2; vaporizer temp = 350 °C.

For accurate metabolite identification, a standard library of ~400 metabolites was analyzed before sample testing and accurate retention time for each standard was recorded. This standard library also forms the basis of a retention time prediction model used to provide putative identification of metabolites not contained within the standard library [72]. Acquired LC-MS/MS data was processed in an untargeted fashion using opensource software IDEOM [73], which initially used msConvert (ProteoWizard) [74] to convert raw LC-MS files to mzXML format and XCMS [75] to pick peaks to convert to peakML files. Mzmatch was subsequently used for sample alignment and filtering [76]. Peak intensities were compared for each prototroph versus the matched no-isolate control using a Wilcox Rank Sum test with Benjamini-Hochberg multiple testing correction. Missing and zero peak intensity data were imputed as 20% of the minimum peak intensity for each prototroph-control pair.

### Data visualization and code availability

Statistical analysis and graphical visualization were performed using R v4.0.3 [77], RStudio v1.3.1093 [78], with the following software packages: tidyverse v1.3.1 [79], viridis v0.5.1 [80], RColorBrewer v1.1-2 [81], ggpubr v0.4.0 [82] ggpmisc v0.4.4 [83], aplot v0.1.6 [84], colorspace v2.0-2 [85], ggtree v2.4.1 [86], vegan v2.5-7 [63], rstatix v0.7.0 [87], and ggnewscale v0.4.5 [88].

All code used to generate metabolic models and simulate growth phenotypes, including media definitions, is available at https://github.com/kelwyres/Bactabolize. The KpSC pan-reference metabolic model is available at https://github.com/kelwyres/KpSC-pan-metabolic-model. All code used to generate figures, simulations, and perform statistical analysis can be found at (https://doi.org/10.6084/m9.figshare.24503737), alongside all metabolic models used in analyses.

## Results

### Metabolic traits are bimodally distributed across the population

The study dataset included 7,835 high-quality KpSC genomes covering all seven taxa in the complex (Fig 1A) and 1,931 STs (S1 Data). Pan-genome analysis [59] identified 61,595 gene clusters, 11,184 (18.2%) of which were matched to a total of 2,601 unique KEGG Orthologs [61] with associated substrate-reaction data. This represented a lower bound estimate of the proportion of 'metabolic genes' in the KpSC pan-genome, as 38.9% of gene clusters could not be assigned functional annotations (hypothetical proteins). Metabolism-associated KEGG Orthologs (hereafter 'metabolic orthologs') were bimodally distributed (Fig 1B and S4 Data), with 1,499 orthologs (57.4%) present in ≥95% of genomes and 899 rare orthologs present in <15% of genomes. This bimodal distribution is consistent with the general distribution of accessory genes in the KpSC [4], and other bacterial species with similarly large effective population sizes [64,89].

To analyze the phenotypic diversity across the KpSC, a strain-specific metabolic model was generated for each genome using Bactabolize [65], which was previously shown to produce models with overall 93.2% and 78.1% predictive accuracies in aerobic and anaerobic conditions, respectively (data for 37 diverse KpSC each tested for growth using 124 distinct carbon substrates via the Omnilog microarray; range 5.4%–100.0%, median 100.0% accuracy per substrate in aerobic conditions; range 0.0%–100.0%, median 88.9% accuracy per substrate in anaerobic conditions [66]). Growth phenotypes were predicted for 345 carbon, 192 nitrogen, 68 phosphorus, and 34 sulfur substrates as sole sources in minimal media under aerobic and anaerobic conditions (a total of 1,278 conditions). Individual strain-specific models comprised a mean of 1677.0 genes ± 17.3 (standard deviation, SD) and 3330.1 reactions ± 13.5 (SD). Individual isolates were predicted to grow in a mean of 678.2 ± 17.9 (SD) conditions. Seventeen isolates were identified as outliers with fewer positive growth phenotypes in anaerobic conditions, likely due to sequencing coverage issues rather than genuine biological deficiencies [90] (S5 Data, S1A Fig, and S1 Text).

The distribution of positive growth predictions mirrored that for metabolic gene content. At the population-level, 643 conditions (47.6%) were predicted to support growth for ≥95% isolates ('core', Fig 1C and S5 Data) and were comprised of 349 aerobic and 294 anaerobic conditions, which systematically confirmed the facultative anaerobe status of the KpSC. These 643 core conditions consisted of 207 total substrates (note that some substrates can be tested as sole sources of multiple elements, e.g., carbon and nitrogen), with at least 204 of these being known human metabolites, 134 diet-related compounds, and 203 gut microbiome metabolites (S5 Data). In contrast, 509 modeled growth conditions (43.6%) did not support growth of any isolates (224 aerobic and 285 anaerobic conditions). Carbon substrate usage displayed the greatest variability, with 41 aerobic and 37 anerobic conditions predicted to support growth in <95% isolates (Figs 1C and S1B).

### Metabolic variation is structured within and between species

The set of core growth conditions for each KpSC taxon was greater than the species complex as a whole (11–39 additional conditions each). Each taxon was associated with a distinct core growth profile mirrored by distinct core metabolic

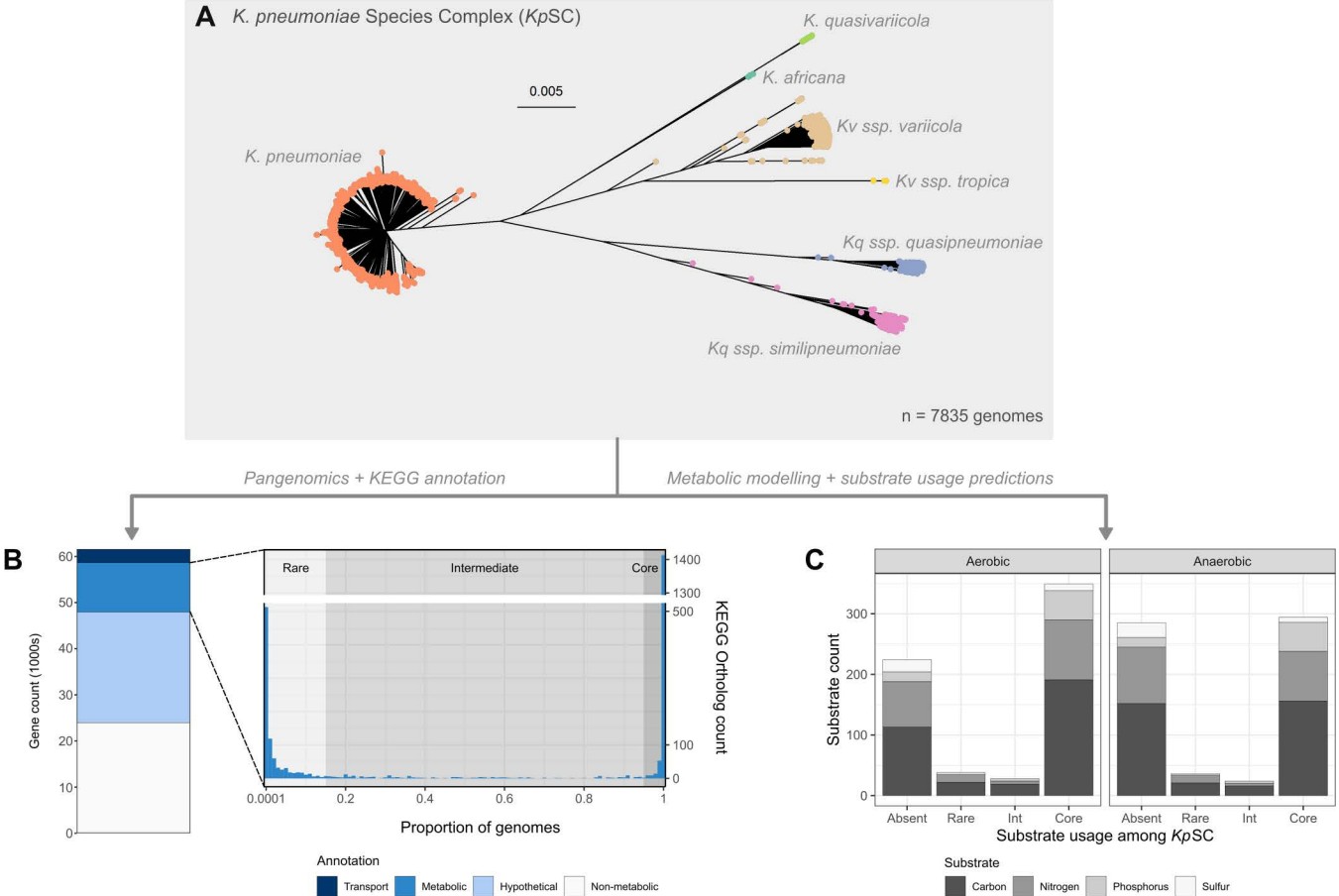

**Fig 1. Population-level metabolic diversity in the *K. pneumoniae* species complex.** A: Neighbor-joining tree of the 7,835 *Kp*SC genomes used in this study. Tip colors indicate species as labeled. An interactive version found at https://microreact.org/project/pcTMwQZAGCsqqhbEZzaj11-a-metabol-ic-atlas-of-the-klebsiella-pneumoniae-species-complex. B: Stacked bars show pan-genome content by metabolic gene (KEGG) category as indicated in legend (left). Population distribution of KEGG orthologs associated with metabolic genes ('metabolic orthologs', right). The light gray background shows rare orthologs (>0–15%), the medium gray shows intermediate orthologs (>15–<95%) while the dark gray shows core orthologs (≥95%). C: Stacked bars show population frequency categories for predicted substrate usage, coloured by substrate type as indicated in legend. The data and phylogeny underlying this Figure can be found at Figshare (https://doi.org/10.6084/m9.figshare.24503737) and S4 Data and S5 Data.

ortholog profiles as expected [70,91] (S2–S4A Figs, S4 Data, S5 Data, and S1 Text), and there was evidence of special-ization at the level of macromolecular functions as determined by COG analysis (S5 Fig, S1 Data, and S1 Text). These predicted growth capabilities were generally consistent with biochemical tests described in formal species definitions [92–95]. However, there were a small number of notable discrepancies that indicate that species identification protocols should be revisited (see S1 Text and S6 Data for details).

While individual *Kp*SC taxa are clearly defined by distinct metabolic profiles, there was also clear evidence of intra-taxon variation (S2 and S3 Figs). Four taxa were represented by sufficient genomes for comparisons ($n > 200$ each); *K. pneumoniae* (hereafter *Kp*, $n = 6,652$), *Klebsiella quasipneumoniae* subsp. *quasipneumoniae* (*Kqq*, $n = 201$), *K. quasipneumoniae* subsp. *similipneumoniae* (*Kqs*, $n = 285$), and *Klebsiella variicola* subsp. *variicola* (*Kvv*, $n = 672$). In *Kp*, 938 metabolic orthologs were variably present in <95% genomes (544, 586, and 738 orthologs variable among *Kqq*, *Kqs*, and *Kvv*, respectively) and 79 conditions were predicted to support variable growth (81, 78, and 67 for *Kqq*, *Kqs*, and *Kvv*). We

PLOS Biology

hypothesized that this variation is not randomly distributed in the population, and our analysis of all-vs-all genome pairs within taxa showed statistically significant positive relationships between pairwise Average Nucleotide Identity (ANI, generally indicative of ancestral relatedness) and pairwise Jaccard similarity across metabolic genes (general linear hypothesis test, $p < 2 \times 10^{-16}$ for each of the four well-sampled taxa, **s 2A**). Comparison of the regression gradients indicated significant differences between taxa; however, the $R^2$ values, which indicate how well the models fit the data, were low (range 0.04–0.48), suggesting a large amount of variation in metabolic gene Jaccard similarities was not explained by the relationship with ANI.

Next, we used the recently established core-genome Life Identifier Number (LIN) code scheme to assign genomes to SLs [54]. These SLs represent deep-branching phylogenetic clades, separated by hundreds of years of evolutionary divergence (with notable exceptions discussed below). Of the 7,835 genomes, 7,833 were assigned to 1,418 distinct SLs and two were unassigned due to insufficient alleles detected.

Pairwise Jaccard similarities for metabolic ortholog profiles were significantly higher within SLs (median 0.956, IQR 0.947–0.965) than between SLs within the same species (median 0.923, IQR 0.914–0.932), $p < 0.0001$ (Kruskal–Wallis test). Mirroring this, pairwise Jaccard similarities for predicted substrate usage profiles were significantly higher for pairs within SLs (median 0.990, IQR 0.983–0.997) than between SLs (median 0.97, IQR 0.96–0.98) for both aerobic and anaerobic substrate usage ($p < 0.0001$ for all comparisons, Kruskal–Wallis test with Holm correction, followed by Dunn's post-hoc, Fig 2B).

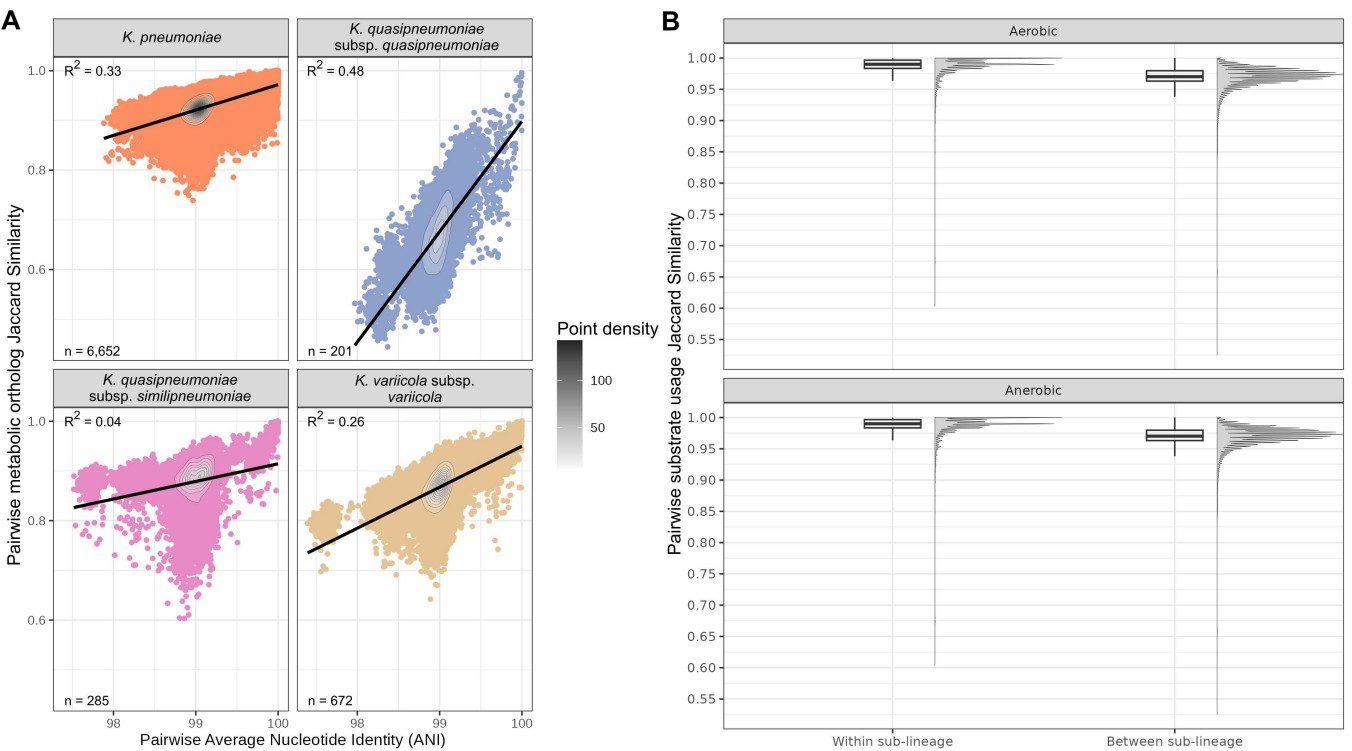

**Fig 2. Metabolic heterogeneity within taxa is nonrandomly distributed.** A: Pairwise Average Nucleotide Identity (ANI) vs. pairwise Jaccard similarity of metabolic orthologs within each of the well-sampled taxa ($n > 200$ genomes). B: Distributions of pairwise (all vs. all) Jaccard Similarity of simulated substrate usage for pairs of genomes within vs. between SLs in the same species. Significance asterisks excluded as each group was significantly different from every other group ($p < 2 \times 10^{-16}$), calculated using a nonparametric Kruskal–Wallis test with Holm correction, followed by Dunn's post-hoc test. The data underlying this Figure can be found at Figshare (https://doi.org/10.6084/m9.figshare.24503737).

Finally, we sought to explore structuring among accessory metabolic traits, using a co-occurrence analysis. Twelve pairs of co-occurring substrate-usages were identified and 11 of these pairs were mechanistically-linked, e.g., the same substrate tested as different element sources ($n=5$ pairs) or substrates for which the catabolic pathways are overlapping ($n=4$ pairs, see S1 Text). Just one co-occurring phenotype pair (formamide and Fe(III)dicitrate as sources of carbon) was not obviously mechanistically-linked, indicating that they may be co-selected or encoded by genes that are physically linked in the genome or on plasmids. Accordingly, we have identified the genes enabling use of formamide and Fe(III)dicitrate as sources of carbon co-located on plasmids in the completed genomes of multiple unrelated strains, e.g., INF008 (SL221), INF044 (SL258), and INF119 (SL17).

## Major SLs are defined by unique metabolic profiles

To explore the association between metabolism and population structure at higher-resolution, we calculated the relative prevalence of predicted growth phenotypes within 48 *Kp* SLs that were each represented by genomes from broad geographies and time-spans (≥15 genomes, 3–12 geographic regions, 5–86 years). These included SLs corresponding to globally-distributed multi-drug resistant (SL14, SL15, SL17, SL29, SL37, SL101, SL147, SL258, and SL307) and hypervirulent clones (SL23, SL25, SL65, and SL86) [3]. The data showed that each SL is associated with a distinct, core metabolic fingerprint, indicating metabolic specialization (Figs 3, S4B, S6, and S5 Data), with 253–316 core aerobic and 198–223 core anaerobic conditions each. Note that in some cases the SL-specific core was smaller than the *Kp* species core reported above using the ≥95% genome threshold (363 aerobic, 307 anaerobic conditions), because most SLs represent far less than 5% of the total *Kp* genome collection. This contradiction highlights the influence of sample size and diversity on core definitions, which should be considered as indicative of the general trends in the population. We tested for the impact of sampling biases on our SL comparisons. The data show no correlation between the number of SL core traits and either of sample size or temporal range (S7A and S7B Fig, Pearson's product moment correlation, $r=0.20$ and 0.04, $p=0.1751$ and 0.7774, respectively). There was a moderate positive correlation between the number of SL core traits and number of geographic regions represented (S7C Fig, $r=0.49$, $p=0.0004$). However, the latter is counter intuitive and seems to be driven by a small number of SLs that have a low number of core traits, rather than a genuine trend (S7C Fig). We are therefore confident that the trends we describe are not the result of sampling biases. Nonetheless, we acknowledge that the precise sets of SL-specific core traits listed here should be considered an approximation, because the addition of further diverse genomes for any given SL could result in the removal of a minority of traits from its core list.

No pair of SLs shared the same core growth profile, even pairs that are thought to share relatively recent common ancestry, such as SL14 and SL15 (common ancestor ~45 years ago [4]) and SL65 and SL25 (common ancestor yet to be dated) [96] (Figs 3 and S6). However, there was also clear evidence of diversity within SLs, with 3–102 variable growth phenotypes each, a pattern that was replicated by metabolic ortholog distributions (S3, S6 Figs, and S1 Text). After excluding growth conditions predicted to support growth among ≥95% (46/48) *K. pneumoniae* SLs ($n=615$ conditions, 340 aerobic and 275 anaerobic, corresponding to 202 total substrates) and those that did not support any growth among the 48 SLs ($n=525$ conditions), the data support a broad classification of phenotypes as follows: i) 89 that were core in some SLs (range = 1–45 SLs) but variably present in others (range = 1–17) which we call 'SL-specific core traits'; ii) 13 that were not core in any SL (accessory in 3–46 SLs, including the only pair of co-occurring phenotypes that was not mechanistically-linked) (Figs 3 and S6). The SL-specific core traits could be further divided into those that were core to the majority (≥66%) of SLs ($n=57$ phenotypes that we call 'majority SL-specific core traits'), those that were core to an intermediate number of SLs (≥25%–<66%, $n=13$ phenotypes, corresponding to six substrates, that we call 'common SL-specific core traits') and those that were core in only very few SLs (<25%, $n=13$ 'rare SL-specific core traits', eight aerobic and five anaerobic, corresponding to seven total substrates).

We have previously tested the predictive accuracy of 16 of 68 variable aerobic growth phenotypes, 14 of which were estimated ≥70%, including 12 estimated ≥90% (see Fig 3), while the accuracy for 11 of 16 tested anaerobic

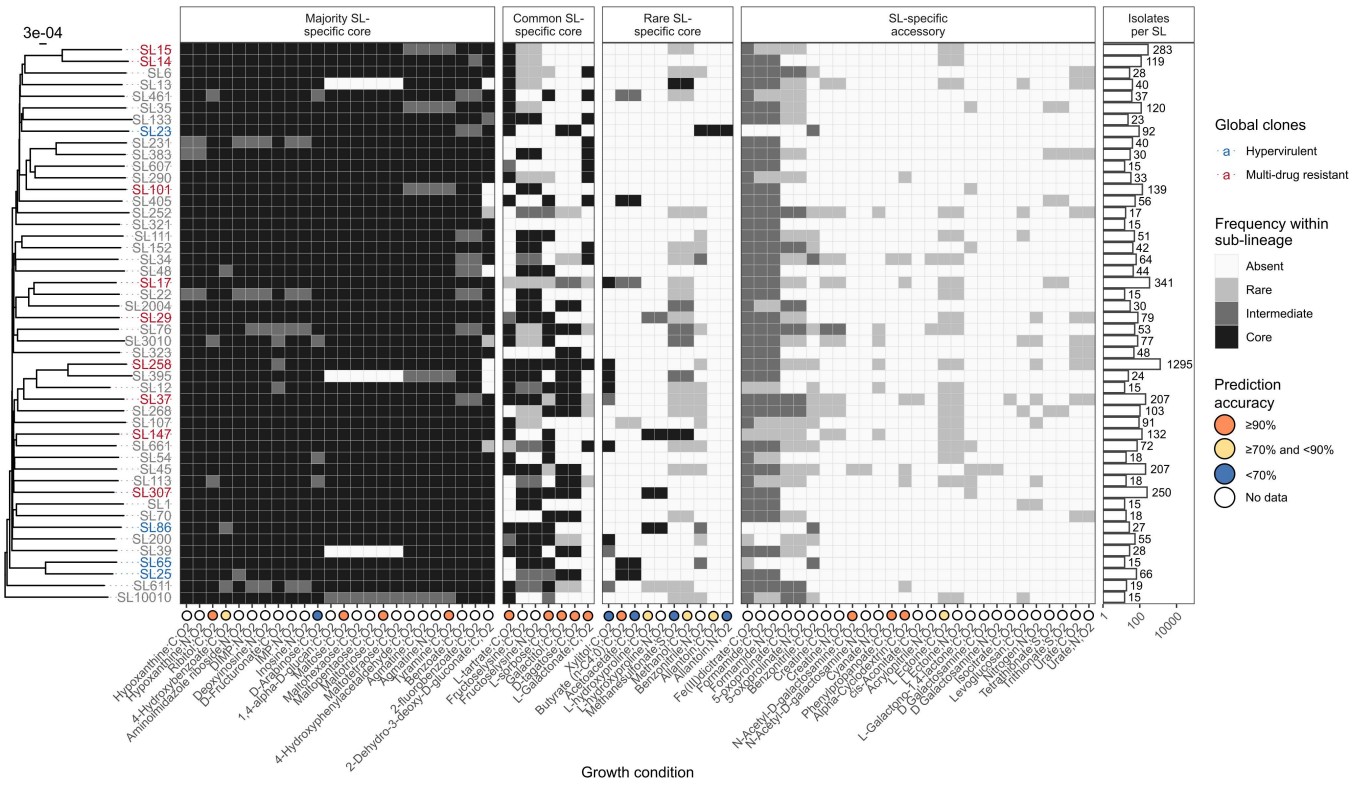

**Fig 3. Distinct metabolic fingerprints of SLs within species.** Heatmap showing frequency of variable aerobic substrate usage across 48 global *K. pneumoniae* sub-lineages (SL). Rows are ordered by phylogeny. SL labels are coloured to indicate the globally-distributed clones described in [3]: Blue shows hypervirulent while red shows multi-drug resistant. Substrate shown along X-axis. The element source of each substrate is semi-colon separated and abbreviated for brevity: C = Carbon, N = Nitrogen, and S = Sulfur. O2 indicates aerobic conditions. Frequency of substrate usage indicated by shading as shown in legend. Where available, substrate-specific prediction accuracies are indicated in coloured circles below the X-axis. Number of isolates per each SL shown in bars. Anaerobic growth predictions shown in S6 Fig. The data underlying this Figure can be found in S5 Data.

growth phenotypes was estimated ≥70%, including three estimated ≥90% (see S6 Fig). We propose that the majority SL-specific core traits and the accessory traits are unlikely to play a major role in defining SL-specific evolutionary trajectories or ecological interactions, and hence we focused our follow-up investigations here on the common and rare SL-specific core traits. Our prior work had confirmed predictive accuracies ≥97% for five of the six common SL-specific core substrates; L-sorbose, L-tartrate, galactitol, D-tagatose, and L-galactonate as sole sources of carbon in aerobic conditions [66] (tested using the Omnilog phenotyping microarray, for 37 diverse *Kp*SC isolates). The sixth, fructose-lysine, is not present in the Omnilog array and could not be tested here due to its high cost. One of the seven rare SL-specific core substrates, allantoin, was previously tested as a sole source of carbon in aerobic conditions, for which the predictive accuracy was 89.2% (37 isolates tested [66]). Here, we tested growth using allantoin as a sole source of nitrogen and found the model accuracy was much lower (15% for 13 isolates tested). We also tested the six other rare SL-specific core growth substrates (13 isolates tested for each), confirming high accuracy (85%) for L-hydroxyproline as a carbon source, 77% accuracy for each of butyrate and methanol as sources of carbon, 46% for xylitol as a source of carbon, 31% for methanesulfonate as source of sulphur, and 15% for acetoacetate as a source of carbon (S2 Data, see S1 Text for more details, Fig 3).

## SLs co-operate through cross-feeding metabolites

We have shown substrate usage varies across the *Kp*SC and some of this variation is SL-dependent. We hypothesized that such diversity promotes the maintenance of distinct SLs by limiting inter-lineage competition and/or promoting commensalism. We tested this hypothesis using in vitro co-culture experiments.

We selected four carbon substrates (galactitol, L-tartrate, L-sorbose, and L-hydroxyproline) for use in three independent substrate combinations and corresponding SL pairs for which predicted substrate-usages were mutually exclusive SL-specific core traits. One isolate from each SL was selected for co-culture aerobically in each of two conditions (minimal media plus each of the SL-specific core substrates), such that each experiment contained one isolate that was able to utilize the substrate as a sole carbon source (the prototroph) and one that was not (the auxotroph). In short, Strain A as an auxotroph and Strain B as a prototroph, after which we then swapped their roles via use of another substrate, so that A would be a prototroph and B would be an auxotroph. This was done within each pair to evaluate reciprocal cross-feeding between SLs under different conditions. Growth of the auxotroph was measured using $OD_{600}$ and compared to the same isolate growing in monoculture in the same media. Co-cultured isolates were physically separated by a Transwell membrane but had shared access to the growth medium (Fig 4A). A unique aspect of this experiment was the use of independent isolates from the same SLs as biological replicates rather than replicates of the same isolate. This allowed us to limit the impact of strain-specific growth biases while demonstrating conserved behavior across a greater number of independent isolates. SLs representing clinically-relevant clones were selected (multi-drug resistant SL37 versus hypervirulent SL29, hypervirulent SL86 versus hypervirulent SL23, and multi-drug resistant SL17 versus SL29).

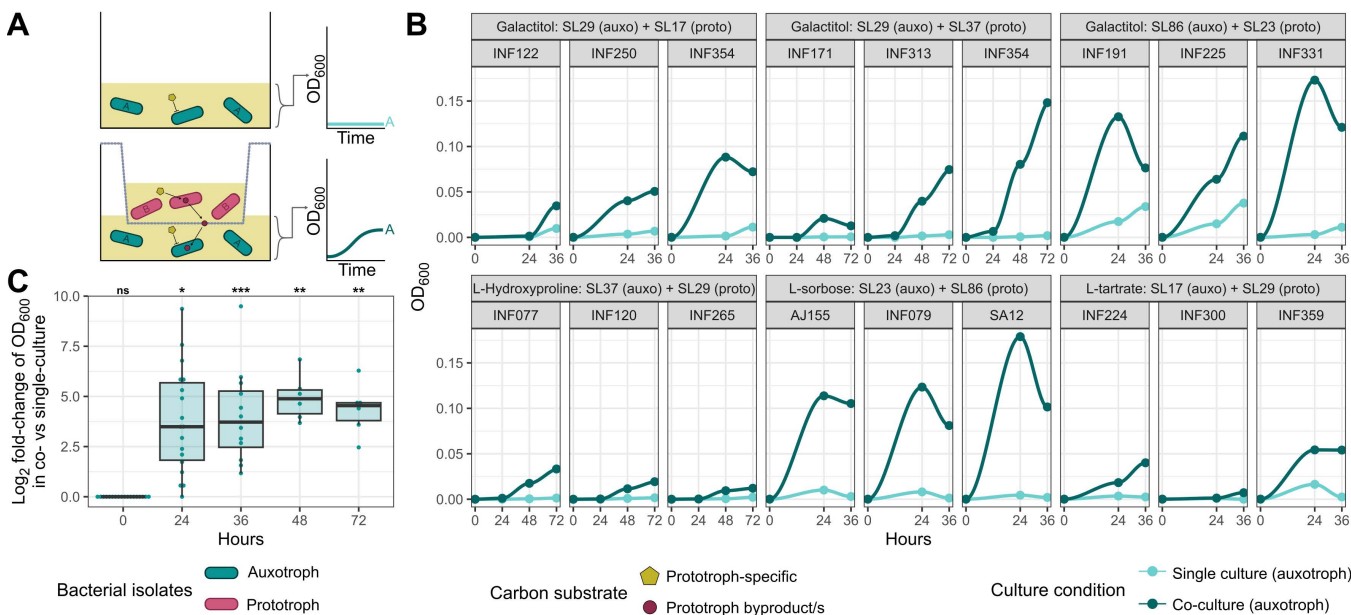

**Fig 4. Nutrient prototrophic lineages support growth of auxotrophic lineages.** Co-culture growth assays between prototrophs and auxotrophs under various substrate usage conditions. Time points captured were at 0, 24, 36, 48, and 72 hours depending upon the substrate pair (Materials and methods). A: Graphic demonstrating the experimental design measuring commensalism between an auxotrophic strain unable to utilize a carbon source and a prototrophic strain which can. The growth of the auxotroph was compared in single vs. co-culture conditions. B: Growth results ($OD_{600}$) over time for each auxotrophic strain in the bottom well (*n* = 3 strains per SL as biological triplicates) in single and co-culture with a prototroph for the substrate. C: Log$_2$-fold-change distribution of growth results ($OD_{600}$) at each time point under different culture conditions. Statistical significance comparing the $OD_{600}$ values of co-culture to single culture is shown above plot (Dunn's post-hoc with Holm correction, *$p$ < 0.05, **$p$ < 0.01, ***$p$ < 0.001). The data underlying this Figure can be found in S3 Data.

Our data showed that co-culturing auxotrophs with prototrophs in the presence of a SL-specific core substrate significantly increased the $OD_{600}$ of the auxotrophs (Fig 4B) with mean log$_2$-fold-change 3.78 ± 2.64 SD at 24 hours, $p = 0.0128$; 4.02 ± 2.34 SD at 36 hours, $p = 0.0001$; 4.94 ± 1.14 SD at 48 hours, $p = 0.0039$; and 4.35 ± 1.27 at 72 hours $p = 0.0039$; Kruskal–Wallis + Dunn's post-hoc with Holm correction (Fig 4C). To confirm this phenomenon only occurred in the presence of a cross-feeding partner and eliminate the possibility of contamination or horizontal transfer of the genes conferring the prototroph phenotype, aliquots from each co-culture well were plated onto LB agar to confirm a single, distinct colony morphology. Colonies were then inoculated into M9 minimal media plus the auxotrophic substrate as a sole carbon source, confirming the original growth behavior of each isolate (S3 Data).

To further explore these pairwise strain interactions, we performed in silico co-culture simulations, which indicated that prototrophs were likely cross-feeding the auxotrophs by catabolising the sole carbon substrates, then exporting metabolites that were capable of supporting auxotroph growth (see S1 Text). Subsequent metabolomics analysis of representative prototroph culture supernatants supported the presence of many of the implicated metabolites (S6 Data, S8 Data, and S8 Fig): Between two and 13 metabolites predicted to be involved in cross-feeding were detected with significantly higher peak intensity in the prototroph culture supernatant compared to the no-isolate controls ($p < 0.05$ by two-tailed Wilcox Rank Sum Test with Benjamini-Hochberg multiple testing correction). Among these, between two and 11 metabolites were each detected with >1 log$_2$ fold intensity increase, indicating a substantial increase in the quantity of the metabolite present in the prototroph supernatant as compared to the matched control (S6 Data). Metabolites with substantial intensity changes included those involved in central carbon metabolism and/or that have been previously shown to support growth for all or almost all *Kp*SC, e.g., alanine, aspartate, glycerate, glutamate, lactate, malate, and pyruvate [70] (labeled in S8 Fig).

We cannot rule out the possibility that the putative cross-feeding metabolites were released following cell lysis; however, we note that the associated peak intensity shifts were among the most prominent of all metabolites tested, including ubiquitous intracellular metabolites, e.g., other amino acids (see S8 Fig). Therefore, it is unlikely that cell lysis alone explains the metabolite quantities we detected. Furthermore, each of the substrates named above is known to be actively exported by the closely related organism *E. coli* [97–102], and a tBLASTn search showed that all six of the representative prototrophs carried orthologs of genes encoding the relevant substrate exporter proteins characterized from *E. coli* (alanine, glutamate, malate, lactate, and pyruvate) or *Tetragenococcus halophila* (aspartate). (We were unable to identify any known exporters of glycerate. Also see S1 Text.)

## Discussion

Here, we present a bacterial population metabolism study of the *K. pneumoniae* species complex, in which we used quantitative metabolic modeling to predict and compare growth phenotypes for >7,000 members of a single species complex, and contextualized the results against an established population genomics framework [3]. This approach allowed us to identify key metabolic differences that are indicative of metabolic fingerprints at both the species and sub-species (sub-lineage) levels. Our subsequent in vitro co-culture experiments (Fig 4) provide evidence that these metabolic differences can facilitate co-operation between SLs.

Our data shows that metabolic capabilities are structured in the population: we observed significant positive correlations between pairwise ANI and metabolic gene Jaccard similarity among each of the four well-sampled *Kp*SC taxa (Fig 2A), indicating that closely related strains (high ANI) generally share a greater overlap of metabolic genes than more distantly related strains (low ANI). However, the proportion of variation explained was low (≤0.48 for all taxa). While there is likely a small impact of unavoidable assembly and/or annotation artifacts, we hypothesize the dominant driver is horizontal gene transfer; i) chromosomal recombination, which can introduce nucleotide variation into core genes and inflate ANI values for otherwise closely related genomes that share common metabolic genes; and ii) movement of mobile genetic elements, driving acquisition or loss of genes among closely related genomes (high ANI) and/or acquisition of common

sets of genes by distantly related genomes (low ANI). This is further supported by the comparison of pairwise growth prediction Jaccard similarities for isolate pairs within versus between SLs (Fig 2B, wherein the distributions were overlapping but shifted towards higher values for within SL pairs) and by our exploration of 48 common SLs (Figs 3 and S6). The latter showed that each SL was associated with a distinct set of core traits (metabolic orthologs and growth phenotypes), alongside clear evidence of variation within SLs, mirroring variations in total gene content that have been reported previously [4,96].

SL-specific core traits likely represent those vertically inherited from the SL common ancestor, whereas variable traits within SLs reflect acquisitions and/or losses through evolutionary time. Interestingly, several predicted growth phenotypes were observed only as accessory traits, i.e., they were not core to any single SL, most likely due to acquisition on mobile genetic elements rather than vertical inheritance followed by repeated losses in diverse strain backgrounds. Accordingly, we identified the genes associated with the two co-occurring accessory growth phenotypes, formamide and Fe(III)dicitrate, co-located on plasmids in genomes representing multiple distinct SLs. A plasmid born lactose utilization operon has been reported previously among *Kp*SC [4], and *E. coli* plasmids have been shown to carry a variety of metabolic genes [103], but the potential role for plasmids in driving metabolic diversity within species has not been fully appreciated.

Our co-culture experiments indicated that metabolic variation, including SL-specific core growth capabilities, can enable co-operation between *Kp*SC wherein an isolate which cannot metabolize a given substrate (an auxotroph), is supported to grow by the breakdown of the substrate by another isolate (a prototroph) (Fig 4). These findings were replicated for each of three distinct isolate pairs, representing three distinct and distantly related SL pair combinations, each tested for two different substrates. In each case, substrates predicted to support growth of the auxotroph were identified within the culture supernatant of one of the prototrophs, and include those ubiquitously metabolized by *Kp*SC. We therefore predict that this capacity for 'cross feeding' is a common phenomenon among *Kp*SC, although we cannot comment on its true frequency in natural environments nor in the context of the broader microbial communities within which *Kp*SC exist. Further work leveraging microbial community models in representative conditions, and/or using in vivo models will be required to explore these intra-species dynamics.

Intriguingly, our data may shed light on the question of how so many SLs can be maintained within the *Kp*SC population. In the most intuitive scenario, the ability to consume different substrates may enable *Kp*SC SLs to inhabit distinct metabolic niches within spatially overlapping environments, thereby reducing the impact of nutrient competition. Supporting this hypothesis, it has been shown previously that metabolic niche differentiation can support the co-existence of multiple strains of *E. coli* in an in vivo model of mouse gut colonization [104]. In a more complex scenario, the ability to consume a substrate that cannot be consumed by a competitor can provide a selective advantage resulting in a relative frequency increase. This in turn can result in a depletion of the food resource and shift in selective pressure to favor the competitor. Although comparatively understudied, commensal interactions such as the metabolite cross-feeding implicated by our co-culture experiments (Fig 4), can also drive similar frequency-dependent selective shifts [106]. Over time, such fluctuations may prevent the exclusion of SLs and facilitate a stable equilibrium of co-existence. Notably, a similar model of negative frequency-dependent selection has been proposed previously as a driver for the maintenance of diverse SLs (entities) within bacterial populations [105,106]. However, this model is notoriously difficult to substantiate [106], requiring a long-term dataset from the same location over time and/or laboratory protocols that are beyond the scope of the current work, and therefore it remains to be confirmed.

Niche range no doubt further complicates the eco-evolutionary dynamics within *Kp* populations, e.g., where SLs are exposed to fluctuating selective pressures due to transient migration between niches, and/or where SLs undergo niche-adaptation. Recent analyses of contemporaneous *Kp* collections have suggested that some SLs may not be equally distributed between different hosts and environmental sources, which may indicate some level of niche-adaptation [107–109]. However, the same studies also identify near-identical strains from multiple sources and a subset of SLs that appear to occur frequently in diverse environments, which is indicative of niche migration. Importantly, the available

evidence suggests that wherever *Kp*SC are sampled there exists substantial SL diversity, and the SLs identified in distinct niches are intermingled throughout the species phylogeny [4,33,43,108,109]. Therefore, while our convenience sample of genomes comprised primarily human-derived isolates (6,465/7,835 genomes), we anticipate that the trends we report regarding SL metabolic differentiation are applicable to the broader population. Notably, the limited genomes from nonhuman niches included here suggest that metabolic traits cluster isolates by SL rather than by niche (S4B and S4C Fig). However, we acknowledge that there may be rarer traits and/or SLs associated with nonhuman niches that we have not captured in our sample, and studies encompassing more diverse isolates are required to robustly test for niche-metabolism associations.

While genome-scale metabolic models are powerful approaches for exploring cellular metabolism, they are not without caveats. Firstly, model construction is reliant on curated databases that are limited by current biochemical knowledge [61], and while our pan-metabolic reference model was built from a large diverse collection of strains [66], it cannot possibly capture all metabolism in the *Kp*SC population. Hence, we are likely underestimating the metabolic diversity within the highly variable *Kp*SC. Secondly, growth prediction accuracies can vary by substrate, and are generally lower for anaerobic conditions for which the metabolic processes are less well understood [66]. While many of the core substrates and common SL-specific core traits have been confirmed with high accuracy in previous work, we have not tested all substrates and our phenotypic testing in this work showed variable accuracy. This is not unexpected, particularly for rarer substrate catabolism, which is comparatively understudied and therefore difficult to model, and/or requires unknown regulatory stimuli. Of note, our assays showed that allantoin nitrogen usage, a trait previously associated with hypervirulent infections [68] and predicted by our models as a SL-specific core trait in the dominant hypervirulent SL23, may in fact be core to *Kp*SC (see S1 Text). These and other inaccurate predictions highlight gaps in our current knowledge that are noted for future investigations, and in some cases may have resulted in an overestimation of the population metabolic diversity. On the other hand, given the limitations to modeling traits involving pathways that are not well studied and/or for which the reaction stoichiometries are not defined; and that metabolic models do not consider the impact of sequence variations within metabolic genes nor variations in gene regulation; our analyses are very likely to represent an underestimate of the true metabolic diversity within and between *Kp*SC species and SLs. Regulatory variations likely play an important role in determining metabolic phenotypes in vivo, and in driving competitive interactions and ecological adaptation, and our metabolic models provide a platform on to which transcriptomic, proteomic and metabolomic data can be integrated and contextualized in the future to further probe diversity.

Notwithstanding the caveats of in silico modeling, our analyses clearly challenge the assumption that *Kp* performs a defined and homogenous metabolic function, and this has important implications for how we understand and model *Kp*SC pathogenicity, and/or microbial communities containing this organism. There is increasing interest in gut microbiota manipulation for prevention of pathogen colonization and subsequent opportunistic infection [110,111], and studies on *Kp* have thus far implicated competitive exclusion via nutrient competition as the major mechanistic driver [111,112]. Notably, the *Kp* SL-specific core traits we identified include those implicated in driving competitive interactions and colonization resistance in the mammalian gut, e.g., fructoselysine and galactitol usage [113,114]. Our work therefore provides further evidence for the importance of including diverse strains in experimental studies of *Kp*SC competitive interactions, colonization dynamics, pathogenicity, and virulence (including hypervirulence), and highlights the need to consider population structure when designing and interpreting metabolic association studies. A comprehensive understanding of the population metabolic diversity will greatly benefit the design of novel control strategies targeting *Kp*, particularly those harnessing microbial competitive interactions or targeting metabolic processes.

## Supporting information

**S1 Text. Description of supplementary results.**
(DOCX)

**S1 Data. List of all 7,835 genomes used for this study along with accessions, metadata, per-genome summary of simulated growth phenotypes, and proportional Clusters of Orthologous Genes (COG) data.**
(XLSX)

**S2 Data. Outcomes of in vitro growth experiments for metabolic model validation.**
(XLSX)

**S3 Data. Raw data and statistical analysis of co-culture experiments.**
(XLSX)

**S4 Data. Metabolic gene (metabolic KEGG Ortholog) presence/absence table for entire 7,835 genome dataset.**
(XLSX)

**S5 Data. Binarised growth predictions from metabolic modeling for entire 7,835 genome dataset.**
(XLSX)

**S6 Data. Metabolites detected by LC-MS of prototroph culture supernatants and comparison of peak intensities to matched no–isolate controls.**
(XLSX)

**S7 Data. Biochemical tests described in formal species definitions and consistency with model predictions.**
(DOCX)

**S8 Data. Substrates predicted to be involved in cross-feeding and LC-MS detection.**
(DOCX)

**S1 Fig. Substrate usage frequencies across *Kp*SC dataset.**
(PDF)

**S2 Fig. Taxon-specific substrate usage.**
(PDF)

**S3 Fig. Taxa are associated with unique metabolic ortholog profiles.**
(PDF)

**S4 Fig. PHATE analysis of metabolic orthologs.**
(PDF)

**S5 Fig. Taxon-specific specialization of macromolecular functions.**
(PDF)

**S6 Fig. Distinct anaerobic metabolic fingerprints of sub-lineages within species.**
(PDF)

**S7 Fig. Relationships between the number of core aerobic phenotypes predicted for each common sub-lineage and sample characteristics.**
(PDF)

**S8 Fig. Metabolite peak intensities in isolate culture supernatants as compared to no-isolate controls in M9 minimal media supplemented with various carbon sources as specified.**
(PDF)

## Acknowledgments

This study used BPA-enabled (Bioplatforms Australia)/NCRIS-enabled (National Collaborative Research Infrastructure Strategy) infrastructure located at the Monash Proteomics and Metabolomics Platform. This work was further supported by Monash eResearch capabilities, including Research Data Storage, Nectar Research Cloud and the MASSIVE M3 HPC facility (www.massive.org.au).

## Author contributions

**Conceptualization:** Ben Vezina, Sylvain Brisse, Jonathan M. Monk, Kathryn E. Holt, Kelly L. Wyres.

**Data curation:** Ben Vezina, Martin Rethoret-Pasty.

**Formal analysis:** Ben Vezina, Helena B. Cooper, Christopher K. Barlow, Kelly L. Wyres.

**Funding acquisition:** Sylvain Brisse, Jonathan M. Monk, Kathryn E. Holt, Kelly L. Wyres.

**Investigation:** Ben Vezina, Helena B. Cooper.

**Methodology:** Ben Vezina, Christopher K. Barlow, Sylvain Brisse, Jonathan M. Monk, Kathryn E. Holt, Kelly L. Wyres.

**Project administration:** Kelly L. Wyres.

**Resources:** Ben Vezina, Sylvain Brisse, Jonathan M. Monk, Kathryn E. Holt, Kelly L. Wyres.

**Supervision:** Sylvain Brisse, Jonathan M. Monk, Kelly L. Wyres.

**Validation:** Ben Vezina.

**Visualization:** Ben Vezina, Kelly L. Wyres.

**Writing – original draft:** Ben Vezina, Kelly L. Wyres.

**Writing – review & editing:** Ben Vezina, Helena B. Cooper, Christopher K. Barlow, Martin Rethoret-Pasty, Sylvain Brisse, Jonathan M. Monk, Kathryn E. Holt, Kelly L. Wyres.

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
