## [Editor Report · Decision Letter 0]

7 Mar 2025

Dear Dr Vezina,

Thank you for submitting your manuscript entitled "A metabolic atlas of the Klebsiella pneumoniae species complex reveals lineage-specific metabolism that supports co-existence of diverse lineages" for consideration as a Research Article by PLOS Biology.

Your manuscript has now been evaluated by the PLOS Biology editorial staff, as well as by an academic editor with relevant expertise, and I am writing to let you know that we would like to send your submission out for external peer review.

Once your full submission is complete, your paper will undergo a series of checks in preparation for peer review. After your manuscript has passed the checks it will be sent out for review. To provide the metadata for your submission, please Login to Editorial Manager (https://www.editorialmanager.com/pbiology) within two working days, i.e. by Mar 09 2025 11:59PM.

Kind regards,

Melissa

Melissa Vazquez Hernandez, Ph.D.

Associate Editor

PLOS Biology

---

## [Decision Letter · Decision Letter 1]

11 Apr 2025

Dear Dr Vezina,

Thank you for your patience while your manuscript "A metabolic atlas of the Klebsiella pneumoniae species complex reveals lineage-specific metabolism that supports co-existence of diverse lineages" was peer-reviewed at PLOS Biology. It has now been evaluated by the PLOS Biology editors, an Academic Editor with relevant expertise, and by four independent reviewers.

In light of the reviews, which you will find at the end of this email, we would like to invite you to revise the work to thoroughly address the reviewers' reports. As you will see below, the reviewers are enthusiastic about the study overall but some suggestions and requests were made that will strengthen the study and shade more light on the mechanism. Reviewer 1 would like more insights into the mechanistic details, e.g. which extracellular metabolites support cross-feeding. The reviewer also gives some suggestions that may improve the models. Reviewer 3 would like you to at least discussed the limitations, including mechanistic insights. However, this reviewer also gives several analytical and experimental suggestions that can enhance the study. Reviewer 4 thinks the study should provide more evidence to support the ecological mechanisms proposed and and the GSMM-derived traits.

We would like to high-light 3 points that should be addressed for further consideration:

1. all four reviewers were concerned that the overall conclusions regarding ecology were too broad and that claims should be toned down and better reflect the data.

2. Additional experiments suggested by the reviewers should be considered, and we strongly encourage that they are performed.

3. Details on the modeling should be provided to allow others to reproduce the results.

Given the extent of revision needed, we cannot make a decision about publication until we have seen the revised manuscript and your response to the reviewers' comments. Your revised manuscript is likely to be sent for further evaluation by all or a subset of the reviewers.

**IMPORTANT - SUBMITTING YOUR REVISION**

*Re-submission Checklist*

*Published Peer Review*

*PLOS Data Policy*

*Blot and Gel Data Policy*

Sincerely,

Melissa

Melissa Vazquez Hernandez, Ph.D.

Associate Editor

PLOS Biology

REVIEWERS' COMMENTS:

Reviewer #1:

This study presents a large-scale metabolic analysis of the Klebsiella pneumoniae species complex. The authors used 7,835 genomes to generate strain-specific genome-scale metabolic models (GEMs) and simulated growth under 1,278 distinct nutrient conditions. They demonstrate that both gene content and predicted metabolic phenotypes are non-randomly structured within the population. Sub-lineages possess distinct "core" metabolic capabilities, some of which enable cross-feeding interactions between lineages. These interactions were validated through in vitro co-culture experiments and in silico simulations. Overall, the study reveals that metabolic differentiation contributes to eco-evolutionary dynamics within KpSC and may inform future therapeutic strategies targeting metabolism. The paper is a really strong and well-executed example of using systems analysis to uncover specific trends to validate. Overall I find the question and approach very timely and exciting. However, I feel there is very little mechanistic detail to explain their main observation (cross-feeding), and lack of details on the modeling makes it hard for anyone else to reproduce the data.

Major comments:

1. The accuracy and robustness of sub-lineage definitions should be addressed more explicitly. Several sub-lineages appear under-sampled relative to others. Can the authors assess whether these groupings remain robust under alternative clustering methods (e.g., based on gene content or metabolic profiles)? What is the impact of uneven sampling on the core trait designations?

2. The co-culture experiments show convincing evidence of auxotroph growth in the presence of prototrophs, but the mechanistic basis remains speculative. Can the authors predict or validate (e.g., via targeted metabolomics or in silico exchange fluxes) which extracellular metabolites support cross-feeding? Are transporters or known secretion systems encoded in prototroph genomes? Were unrelated sub-lineages or nutrient sources, or heat-killed controls, tested to rule out nonspecific effects? While the focus is on commensalism, did the authors assess whether prototrophs suffer any fitness cost when supporting auxotrophs? This distinction is crucial for determining whether the interaction is commensal, mutualistic, or parasitic. Longitudinal co-culture dynamics or competitive fitness assays would help clarify this.

3. The manuscript would benefit from greater transparency in strain-specific model construction. Were any manual curation steps performed on the draft GEMs? Can the authors provide SBML, Excel, or JSON versions of representative models (e.g., one per major sub-lineage), including details of media composition and constraints used in flux simulations? Studies like these are always challenging to reproduce since different groups use different standards for flux bound estimates.

4. The low R² values for the ANI-Jaccard similarity analyses suggest that metabolic phenotype divergence is not fully explained by core genome similarity. Could the authors provide more context for their interpretation here? Could the extent to which unannotated accessory genes, variation in regulatory architecture, or mobile elements contribute to this unexplained variance? Why else do they expect it to be so low?

5. While LIN codes offer hierarchical resolution, recombination may erode phylogenetic signal over time. Have the authors considered how homologous recombination affects sub-lineage metabolic coherence and the interpretation of trait inheritance?

6. The models are stated to have high predictive power, but accuracy varies by substrate, particularly for anaerobic and rare nutrient conditions. Would incorporating confidence scores or ensemble modeling approaches help quantify prediction uncertainty, especially for less-characterized conditions?

7. Several figures (e.g., Fig. 1C) are difficult to interpret due to the large number of conditions displayed in stacked bar charts. Consider alternative visualizations such as a binary heatmap or dimensionality reduction (e.g., PCA or UMAP) to depict strain-by-substrate growth predictions and their correlation with lineage structure.

Reviewer #2:

In this manuscript "A metabolic atlas of the Klebsiella pneumoniae species complex reveals lineage-specific metabolism that supports co-existence of diverse lineages" Vezina and co-authors describe an ambitious population-level analysis of over 7000 genomes representing the 7 taxa comprised by the KpSC. The complex harbors hundreds of diverse sub-lineages with non-random gene content and differential epidemiological behavior. Understanding the processes whereby diversity arises and is maintained in pathogen populations is pivotal for designing disease control interventions. Vezina and co-authors specifically sought to evaluate the distribution and diversity of metabolic traits in the context of the population structure and further aimed to interrogate whether metabolic differentiation underlies the existence and maintenance of co-circulating sub-lineages within the population.

The work combines comparative genomics and quantitative metabolic modeling with a selection of in vitro experiments that yield substantiating empirical evidence for reciprocal cross-feeding between sub-lineages. The scale of the analyses allowed these authors to identify key metabolic differences which appear to indicate metabolic fingerprints on species- and subspecies levels. In my view this is an excellent study. Despite the ambitious scope of this project the authors' description is succinct and quite clear. I accept the rationale behind selections made at successive stages of the analysis. While there are caveats to genome-scale metabolic modeling at this time, the authors are well aware of it. Current limitations are acknowledged in the manuscript and I think they are acceptable. The cooperative co-culture data obtained are technically sound and supportive.

This research makes a very nice contribution. The potential for tailoring Klebsiella pneumoniae control strategies to metabolic processes and dependencies revealed by this study is certainly intriguing. I have no major criticism.

Reviewer #3 (Guido Werner):

The study provides a large-scale, systematic analysis of metabolic diversity within the Klebsiella pneumoniae species complex (KpSC). By combining comparative genomics with high-throughput metabolic modeling, the authors introduce new insights into lineage-specific metabolism and its role in bacterial ecology and evolution. The study employs genome-scale metabolic modeling, a powerful computational approach, to predict metabolic traits across 7,835 genomes. The authors validate a subset of predictions using experimental assays, strengthening the credibility of their findings. The study successfully links metabolic variation with phylogenetic structure, demonstrating that metabolic profiles are non-randomly distributed across sub-lineages. This contributes to our understanding of bacterial adaptation and niche specialization. The in vitro co-culture experiments effectively confirm the existence of reciprocal cross-feeding interactions between sub-lineages, adding empirical support to the computational predictions. The findings have implications for developing new antimicrobial strategies and microbiome-based interventions, particularly in preventing K. pneumoniae colonization and infection.

I identified some weaknesses/areas for improvement which should at least be considered and discussed in a Limitations paragraph in the Discussion:

1. Incomplete Experimental Validation: While the authors validate some metabolic predictions, only a subset of the proposed lineage-specific metabolic traits are tested. Additional in vitro and in vivo studies would strengthen the conclusions.

2. Potential Bias in Sample Selection: The dataset primarily consists of human-derived isolates (~83% of the total genomes), which may limit generalizability to environmental and animal-associated strains.

3. Limited Functional Analysis of Key Metabolic Pathways: While metabolic variations are described at a population level, deeper functional studies on key pathways (e.g., transcriptomics, enzyme activity assays) would enhance mechanistic understanding.

4. Challenges in Interpreting Model Accuracy: The metabolic models rely on databases with inherent gaps in knowledge, leading to potential inaccuracies in predicting rare metabolic traits. The reported discrepancies between predicted and observed phenotypes highlight the need for further refinement.

5. Lack of Direct Clinical Correlations: Although the study discusses potential clinical implications, it does not explicitly analyze the relationship between metabolic traits and clinical outcomes, such as infection severity or treatment resistance.

6. Limited study design: While the authors documented a cross-feeding effect across KpSC lineages, the real world scenario includes thousands of commensal bacteria that may cross-feed KpSC strains across species, genus, family etc barriers under natural conditions, for instance, in the human or animal intestinal tracts.

Minor comments

1. Limited Subset of Validated Traits

o Line 351-359 (Page 9): The manuscript states that five of six common sub-lineage-specific core substrates were previously confirmed, but only seven of the rare sub-lineage-specific core growth substrates were tested in this study. It would be beneficial to clarify why these specific substrates were chosen and whether further validation is planned.

o Suggested Improvement: Include a rationale for substrate selection and discuss whether additional phenotypic tests are planned to validate other metabolic traits.

2. Limited Scope of Co-Culture Experiments

o Lines 386-413 (Page 10-11, Figure 4): The co-culture experiments demonstrate cross-feeding between selected sub-lineages, but only a limited number of strain pairs and substrates were tested.

o Suggested Improvement: Explain whether these findings can be generalized across other K. pneumoniae sub-lineages and whether additional co-culture experiments are planned.

Methodological Clarifications Needed

1. Accuracy of In Silico Predictions

o Lines 220-226 (Page 6, Table S5): The study states that metabolic models were 93.2% accurate for aerobic and 78.1% accurate for anaerobic growth phenotypes.

o Suggested Improvement: Explain the limitations of metabolic models in anaerobic conditions and provide possible strategies for improving accuracy (e.g., integrating transcriptomics or enzyme activity assays).

2. Sample Bias in Genome Selection

o Lines 458-460 (Page 12): The manuscript states that 6,465 out of 7,835 genomes (~83%) are human-derived, potentially limiting ecological generalizability.

o Suggested Improvement: Discuss the potential impact of this bias on the study's conclusions and suggest future studies including more environmental and animal-derived isolates.

Lines 251-260: Please add 1 to 2 examples in the main text where genome data support misidentification of Klebsiella species only based on biochemical ID usage patterns.

Lines 273-275: Please explain in more details what a low R2 value means and why this indicates a "large amount of unexplained variation".

Recommendation: Overall, the manuscript presents a well-executed and scientifically valuable study that advances our understanding of K. pneumoniae metabolism and its role in bacterial population structure. Despite some limitations, the findings are compelling and have broad implications for microbial ecology, evolution, and antimicrobial resistance. With minor revisions to address experimental validation gaps and clarify certain methodological aspects, this study should be suitable for publication in a high-impact journal.

Reviewer #4:

Overall Assessment:

Vezina et al. present an extensive computational analysis of the Klebsiella pneumoniae species complex (KpSC), leveraging pangenomics and genome-scale metabolic modelling (GSMM) across 7,835 genomes, complemented by in vitro experiments. Their central claims are that KpSC sub-lineages exhibit distinct metabolic fingerprints ("sub-lineage-specific core traits") and that observed in vitro cross-feeding indicates metabolic differentiation promotes sub-lineage co-existence via mechanisms like negative frequency-dependent selection (NFDS). While the dataset scale and methodological integration are significant, the study's conclusions suffer from several critical limitations pertaining to model fidelity, interpretation of experimental results, dataset bias, and the inferential leap made regarding ecological mechanisms. The work provides a valuable resource but requires substantial tempering of its core ecological conclusions.

The study reports potentially important findings and a valuable dataset. However, the link between the computational predictions, limited experimental data, and the major ecological conclusions requires significant strengthening and more cautious interpretation to be suitable for publication in PLOS Biology. Addressing the concerns regarding the robustness of the GSMM-derived traits and the evidence for the proposed ecological mechanisms is critical for publication. I would recommend a revision of the manuscript in this regard.

Critique:

1. Novelty and Significance:

The scale of the comparative GSMM analysis (>7,800 genomes) linked to fine-scale population structure (sub-lineages) is novel and commendable. The study generates a valuable resource (models, phenotype predictions) and presents intriguing findings regarding sub-lineage-specific metabolic potential and in vitro cross-feeding. The novelty primarily lies in the scale of GSMM application to KpSC sub-lineages, rather than the general concept of metabolic niche partitioning in bacteria. The significance of identifying "sub-lineage-specific core traits" is potentially high but is currently undermined by methodological uncertainties (see below). The link established between these traits and ecologically relevant outcomes like stable co-existence is tenuous. The implications for understanding KpSC co-existence and potentially informing control strategies would be highly significant if the robustness of key In Silico findings was established.

2. Methodology and Interpretation - GSMM Limitations:

o Sensitivity of "Specific Core Traits": The identification of these differentiating traits is highly sensitive to the inherent limitations of the GSMM approach used. Reliance on incomplete annotations (e.g., KEGG, with ~39% hypothetical genes ) means potentially crucial pathways may be missed or mis-assigned. The lack of regulatory networks, kinetics, and accurate prediction for rarer substrates (evidenced by the allantoin discrepancy ) severely limits confidence that the predicted "fingerprints" accurately reflect functional differentiation under biologically relevant conditions. A few erroneous predictions could dramatically alter the perceived specificity. Addressing this would require: expanded high-throughput phenotypic validation specifically targeting predicted differentiating traits across multiple isolates per sub-lineage, sensitivity analyses of the models, and potentially incorporating transcriptomic constraints or probabilistic modelling.

o Weak Phylogeny-Metabolism Link: The weak correlation reported between core-genome ANI and metabolic gene Jaccard similarity (Fig 2A, low R² values ) suggests that vertical inheritance within sub-lineages explains only a minor fraction of metabolic potential variation. This challenges the notion of deeply rooted, stable metabolic strategies defining sub-lineages and might point towards more dynamic horizontal gene transfer shaping metabolic capabilities.

3. Methodology and Interpretation - Experimental Limitations:

o Cross-Feeding Extrapolation: Demonstrating increased auxotroph growth in a Transwell co-culture system is interesting but provides insufficient evidence for its role in vivo. The in vitro system lacks host factors, microbiota complexity, spatial structure, and flow dynamics. The biological significance of the observed growth increase is unclear, and whether this specific type of commensalism is a major factor in complex environments is questionable. The selection of specific substrate/strain pairs may also have biased results towards finding interactions.

4. Critique of the NFDS Hypothesis:

o Premature Conclusion: The assertion that observed metabolic differentiation and in vitro commensalism support NFDS as a driving force for co-existence is a significant overstatement based on the data provided. NFDS requires demonstration of frequency-dependent fitness effects (an advantage when rare), which this study does not provide. Commensalism could contribute, but is not sufficient evidence on its own.

o Required Data for NFDS: Substantiating metabolically-driven NFDS requires substantially different data, such as: (i) Longitudinal studies tracking sub-lineage frequencies and resource availability in a shared environment; (ii) Experimental evolution demonstrating stable co-existence or frequency oscillations linked to metabolic capabilities; (iii) Direct competition assays showing an inverse relationship between frequency and fitness under relevant limiting conditions; (iv) In situ functional genomics/metabolomics linking pathway activity to abundance; (v) Population genomic analyses seeking signatures of balancing selection on the differentiating metabolic genes.

5. Ecological Relevance and Dataset Bias:

o Clinical Skew: The overwhelming bias towards human clinical isolates (~82% ) severely restricts the generalizability of the findings. The identified metabolic patterns likely reflect adaptations to human host niches and infection pressures, not necessarily the broader ecological strategies of KpSC across its diverse habitats (environmental, non-human hosts).

o Host/Niche Context: The study lacks integration of host metabolic context or interaction with other microbiota members. Fitness in vivo is shaped by host nutrient provision, immune factors, and competition/cooperation within complex microbial communities. Extrapolating from minimal media simulations and simple in vitro tests to these complex scenarios is problematic. Addressing this requires: Niche-stratified sampling and comparative analysis, context-specific modelling using media mimicking in vivo conditions, integration with host/metagenomic cohort data, and validation in relevant in vivo or complex ex vivo models.

6. Technical Soundness: The core computational pipelines appears technically sound. However, significant concerns arise regarding the interpretation and robustness of conclusions derived from the GSMMs due to acknowledged (but perhaps underemphasized) limitations like annotation gaps, predictive inaccuracies for key differentiating traits, and the absence of regulatory dynamics. Furthermore, the link drawn between the in vitro cross-feeding results and the complex ecological mechanism of negative frequency-dependent selection (NFDS) is currently speculative and lacks sufficient direct evidence. The strong bias towards clinical isolates also limits the generalizability of the ecological conclusions.

Required Revisions

Major revisions are necessary to address these points and strengthen the manuscript.

Strengthen Evidence for Ecological Mechanisms: The claims regarding NFDS must be substantially tempered or supported by more direct evidence. This might involve new analyses (if possible with existing data, e.g., population genomic scans for balancing selection on key metabolic genes) or, more likely, a significant reframing of the discussion to present NFDS as a hypothesis suggested by, but not proven by, the current data. Explicitly discussing alternative ecological models would also be beneficial.

1. Address GSMM Limitations & Validate Key Predictions: The authors need to more critically assess and discuss how model inaccuracies and limitations impact the identification and interpretation of "sub-lineage-specific core traits." Ideally, expanded experimental validation focusing specifically on these predicted differentiating traits across more isolates/conditions, or computational sensitivity analyses, would be needed to bolster confidence in these core findings.

2. Contextualize Findings Regarding Data Bias: A more nuanced discussion is required regarding how the clinical isolate bias likely shapes the observed metabolic landscape and limits extrapolation to other environments. Explicitly framing the findings in the context of adaptation to human/clinical niches would be more accurate.

3. Refine Interpretation: Overall, the manuscript needs careful revision to ensure conclusions, particularly ecological ones, do not overreach the presented data.

---

## [Editor Report · Decision Letter 2]

12 Nov 2025

Dear Ben,

Thank you for your patience while we considered your revised manuscript "A metabolic atlas of the Klebsiella pneumoniae species complex reveals lineage-specific metabolism and capacity for intra-species co-operation" for publication as a Research Article at PLOS Biology. This revised version of your manuscript has been evaluated by the PLOS Biology editors, and the Academic Editor.

Based on our Academic Editor's assessment of your revision, we are likely to accept this manuscript for publication, provided you satisfactorily address the remaining editorial points. Please also make sure to address the following data and other policy-related requests.

1) Please add the link to the funding agencies in the Financial Disclosure statement in the manuscript details.

2) Please add to your Competing Interests the following statement “KH is a member of PLOS Biology’s Editorial Board. The other authors declare that no competing interests exist."

3) We do not have a word limit. Please move the supplemental methods, results and references, to the main text which can provide the readers an easier access to all information.

Please supply the numerical values either in the a supplementary file or as a permanent DOI’d deposition for the following figures:

Figure 1BC, 2AB, 3, 4BC, S1AB, S4, S5, S6, S7, S8

5) Please cite the location of the data clearly in all relevant main and supplementary Figure legends, e.g. “The data underlying this Figure can be found in S1 Data” or “The data underlying this Figure can be found in https://doi.org/10.5281/zenodo.XXXXX”

6) Supplementary files (e.g., excel). Please ensure that all data files are uploaded as 'Supporting Information' and are invariably referred to (in the manuscript, figure legends, and the Description field when uploading your files) using the following format verbatim: S1 Data, S2 Data, etc. Multiple panels of a single or even several figures can be included as multiple sheets in one excel file that is saved using exactly the following convention: S1_Data.xlsx (using an underscore).

7) Please provide the tree files for the phylogenetic trees in Figures 1A, S2, S6. Please make sure all relevant figures have scale bars.

8) Please provide the raw metabolomics data and chromatograms for any figures containing metabolomics data by uploading them in a repository like Zenodo, or metabolomic-specific repositories like Metabolomics Workbeck. You can read about our policies and recommendations for repositories here: https://journals.plos.org/plosbiology/s/recommended-repositories

9) Please ensure that your Data Statement in the submission system accurately describes where your data can be found and is in final format, as it will be published as written there

We expect to receive your revised manuscript within two weeks.

*Published Peer Review History*

*Press*

Sincerely,

Melissa

Melissa Vazquez Hernandez, Ph.D.

Associate Editor

PLOS Biology

---

## [Editor Report · Decision Letter 3]

26 Nov 2025

Dear Ben,

Thank you for the submission of your revised Research Article "A metabolic atlas of the Klebsiella pneumoniae species complex reveals lineage-specific metabolism and capacity for intra-species co-operation" for publication in PLOS Biology. On behalf of my colleagues and the Academic Editor, Sebastian E. Winter, I am pleased to say that we can in principle accept your manuscript for publication, provided you address any remaining formatting and reporting issues. These will be detailed in an email you should receive within 2-3 business days from our colleagues in the journal operations team; no action is required from you until then. Please note that we will not be able to formally accept your manuscript and schedule it for publication until you have completed any requested changes.

PRESS

Sincerely, 

Melissa

Melissa Vazquez Hernandez, Ph.D., Ph.D.

Associate Editor

PLOS Biology
